# High Electromagnetic Shielding Effect of Carbon Nanotubes/Waterborne Polyurethane Composites Prepared by “Break-Adsorption” Method

**DOI:** 10.3390/ma15186430

**Published:** 2022-09-16

**Authors:** Yasen Li, Yudong Shang, Mingyue Li, Xiang Zhang, Jiangping He

**Affiliations:** 1School of Textile Science and Engineering, Xi’an Polytechnic University, Xi’an 710048, China; 2School of Materials Science and Engineering, Northwestern Polytechnical University, Xi’an 710072, China; 3State Key Laboratory of Intelligent Textile Material and Products, Xi’an Polytechnic University, Xi’an 710048, China

**Keywords:** carbon nanotubes, waterborne polyurethane, composites, electromagnetic shielding

## Abstract

In recent years, conductive polymer composites have been widely studied for their electrical conductivity and electromagnetic shielding effects due to their advantages of light weight, simple preparation methods, and structural design versatility. In this study, oxidized multi-walled carbon nanotubes/waterborne polyurethane composites (OCNT/WPU) were prepared by grafting oxidized carbon nanotubes onto polyurethane molecular chains through in situ polymerization, using environmentally friendly waterborne polyurethane as the polymer matrix. Then, the OCNT/WPU structure was broken by high shear force, and the loading of CNTs was increased by adsorption, and a new composite structure was designed (denoted by OCWPU). The structure and morphology of OCNT/WPU and OCWPU were characterized by FT-IR and SEM. The structure and morphology of OCWPU with different multi-walled carbon nanotube loadings (CNTs/OCWPU) were characterized by SEM, Raman. Finally, the electrical conductivity and the electromagnetic shielding properties of the composites were investigated. It was found that after application of high shear force, the structure of OCWPU was disrupted and the surface activity of the material increased. With the increase in CNTs content, CNTs formed a rosette structure in the polyurethane matrix and covered the surface, and its electromagnetic shielding effect in X-bond (8.2–12.4 Ghz) would be able to reach 23 dB at 5% CNTs/OCWPU and 66.5 dB at 50% CNTs/OCWPU to meet the commercial needs. With 50% CNTs/OCWPU, an electrical conductivity of 5.1 S/cm could be achieved. This work provides a novel idea for the structural design of conductive polymer composites, which can achieve greater performance with the same carbon nanotube content.

## 1. Introduction

With the development of society and the continuous progress of technology, the emergence of electronic devices and electronic communication has provided great convenience our lives, but the electromagnetic radiation generated by electronic devices will produce a certain degree of damage to the body’s cells in excess of the safe radiation dose, which is detrimental to human health [1,2,3,4,5,6,7,8]. In the United Nations Conference on Human Environment [9], electromagnetic radiation has been listed as one of the main pollutants that must be controlled, so the reduction of electromagnetic pollution will be an important topic for a long time.

Since the 1940s, ferromagnetic materials and metallic materials have been widely used in the field of EMI shielding [10], mainly in the form of material grounding and airtight cavities to reflect electromagnetic waves out, or in the form of electricity or eddy current to generate losses, while achieving the effect of electromagnetic shielding [11,12,13]. Due to the density of metal materials, they cannot be processed, and easy to corrosion problems limit their development and application, and conductive polymer materials have the advantages of light mass, corrosion resistance, low cost and good processing performance, so researchers are widely studied to replace metals and other electromagnetic shielding materials [14,15,16,17,18]. At present, conductive polymer composites (CPCs) have become a promising electromagnetic shielding material.

CPCs are mainly composed of a high conductivity conductive filler and an insulating polymer matrix, where the conductive filler provides carriers that migrate through the polymer matrix through the interaction between the conductive fillers [19,20,21,22,23]. Inorganic materials with electrical conductivity are dispersed into the polymer matrix, and these materials form tendons or network-like pathways to make the composite material conductive. At the same time, when the composite material is exposed to external electromagnetic radiation, according to the electromagnetic shielding mechanism [24], part of the electromagnetic wave incident to the surface of the electromagnetic shielding material, due to the impedance of the material and the impedance of the electromagnetic wave propagation medium does not match, is reflected. The electromagnetic wave is not reflected into the material internal propagation, after multiple reflections and absorptions the electromagnetic wave gradually decays, and finally only a small amount of electromagnetic wave comes through the other side of the material. Therefore, in CPCs, whether the conductive filler builds a complete conductive pathway in the polymer matrix often determines the performance of the composite.

The size and structure of conductive fillers and their fundamental properties are critical to the final electrical properties [25]. Benefiting from their high aspect ratio, excellent electrical properties, excellent mechanical properties and chemical stability, carbon nanotubes are widely investigated for the fabrication of conductive composites. In most cases, CNTs are composed of one or more carbon nanotubes. In the majority of CPCs, the CNTs are covered by an insulating polymer matrix. As a result, the overall conductivity of the composites remains at a rather low level.

Numerous researchers have opened many possibilities in the study of the composite of carbon nanomaterials and polymers [26,27,28,29,30], the application of similar materials in different fields, and the composite of different carbon materials with a variety of polymer modifications, being just two. To improve the electrical conductivity and overall performance of CPC, various methods have been developed, including the incorporation of more CNTs into the composites, the use of surfactant-assisted dispersion, ultrasonic dispersion, freeze-drying, and vacuum filtration to improve the interfacial interaction between the conductive filler and the polymer matrix, and thus control the morphology of the conductive network. Dai Mengwei etl. [31] used a high-pressure microfluidizer to prepare an aqueous dispersion of carbon nanotubes and graphene, with PVP as an auxiliary dispersant, and finally mixed it with WPU. The carbon nanotubes and graphene were extremely dispersed by a high pressure microfluidizer, and finally the low reflectivity composite was obtained. The electromagnetic shielding effect of the composite material with 5% content can reach 30 dB, which can shield 99.9% of electromagnetic radiation. Li. H et al. [32] mixed carbon nanotube dispersions with WPU and vacuum filtered them through a cellulose filter membrane to obtain composite films with isolated structures, and the 10% loading of carbon nanotubes could produce excellent electromagnetic interference shielding effectiveness of 24.7 dB at X-band. Subhashis Sit et al. [33] prepared carbon nanotube dispersions and mixed them with thylene methyl acrylate (EMA) and then obtained composites by coating and drying followed by homogenization and hot pressing. At functionalized multiwalled carbon nanotubes (FMWCNT) loading of 10 wt%, FMWCNT/EMA composites showed EMI shielding effectiveness of 25.1 dB and excellent mechanical and electrical properties. Zhao.G et al. [34] prepared dendritic polyurethane fibers with the help of turbulent forces from a high shear force emulsifier and increased the loading of carbon nanotubes up to 80%, with an electromagnetic shielding effect of up to 60 dB and an electrical conductivity of 25 S/cm using plant roots adsorbing a large amount of soil as a model.

Overall, compared with the traditional method of solution mixing, the design of the isolated structure allows carbon nanotubes to form a complete conductive pathway in the polymer matrix, and the porous structure prepared by the freeze-drying method allows more electromagnetic losses inside the composite, which is a new breakthrough in improving both the conductivity and electromagnetic shielding properties of the composite. The irregular structure designed by the bionic method is another interesting research direction for conductive polymer composites, with polymer morphology design as the focus of research to improve the performance of the composites.

In CPCs, there are various choices of polymer matrix. Among these, waterborne polyurethane is known as an environmentally friendly material and has good film-forming properties, which is in line with the general contemporary principle of green sustainability. In this work, waterborne polyurethane prepolymer was prepared from Poly-tetramethylene-ether-glycol (PTMG) and Isophorone diisocyanate (IPDI), and oxidized multi-walled carbon nanotubes/waterborne polyurethane composites (OCNT/WPU) was prepared by grafting carbon nanotubes with oxygen-containing functional groups into the polyurethane system at the prepolymer stage. Subsequently, inspired by the dendritic polyurethane fiber structure, the OCNT/WPU composite emulsion was “broken” to expose the reactive groups of the composite as well as OCNT. Then, more CNTs were “adsorbed” into the broken polyurethane structure by the “break”-“adsorption” method to obtain composites with different CNTs contents. This new structural design provides a new design idea for the development of the structure of conductive composites. The obtained composites can break the electromagnetic shielding effect of 66.5 dB at 50% CNTs content, and 23 dB at 5% content.

## 2. Materials and Methods

### 2.1. Materials

The pristine multi-walled carbon nanotubes with an average length of 20μm and an average diameter of 4 nm were purchased from Chengdu Organic Chemical Co., Ltd. (Chengdu, China). H_2_SO_4_(98%) was supplied by Sinopharm Chemical Reagent Co., Ltd. (Shanghai, China). Poly-tetramethylene-ether-glycol (PTMG, Mn = 2000) was provided by Jiangyin Jinhai Polymer Materials Co., Ltd. (Jiangyin, China), Jiangsu Province. Sodium nitrate (NaNO_3_) was provided by Damao Chemical Reagent Factory, Tianjin. Sodium dodecyl benzene sulfonate (SDBS), triethylamine (TEA), Isophorone diisocyanate (IPDI), and dimethylol butanoic acid (DMBA) were provided by Bayer (China) Co., Ltd. (Shanghai, China), and Shanghai Yien Chemical Technology Co., Ltd. (Shanghai, China), respectively.

### 2.2. Fabrication of OCNT/WPU Composite Emulsion

First, oxidized multi-walled carbon nanotubes (OCNT) were prepared, after functionalization of the pristine multi-walled carbon nanotubes with oxygen-containing functional groups using mixed acids. We took 2.5 g of pristine multi-walled carbon nanotubes (CNTs), 50 mL of H_2_SO_4_, 5 g of NaNO_3_ into a three-necked flask and heated the three-necked flask in a water bath to 60 °C. Then, the solution was kept at a constant temperature of 60 °C, stirred with a magnetic stirrer at 800 r/min for 2 h. This was followed by a centrifuge at 8000 r/min for 5 min to process the solution; we poured out the acid solution and repeated this operation 4 times until there was no excess acid solution in the centrifuge tube. We added the deionized water into the ultrasonic disperser for 30 min, and filtered the dispersed liquid with a vacuum pump until the pH of the filtrate was neutral. The filtered product was freeze-dried and finally ground into a powder to obtain OCNT.

Then, OCNT/WPU composite emulsion was prepared. 20 g of PTMG and 0.9778 g of DMBA were added to a three-neck flask, heated to 110 °C (±5 °C), and evacuated for 2 h to remove the water from the system; the system was cooled down to 60 °C, the vacuum pump was removed, the stirring rod was replaced, and 4.45 g of IPDI was added to the flask. After that, the system was warmed up to 80 °C, maintained the speed of 300 rpm/min and stirred for two hours until the system reached the theoretical reaction value. The system was cooled down to 70 °C, 0.2542 g OCNT was added and stirred for 1.5 h. After 30 min reaction, 86 mL deionized water was added and emulsified with high-speed stirring to obtain 1% OCNT content of OCNT/WPU.

### 2.3. Fabrication of CNTs/OCWPU composites

Took 4 g of OCNT/WPU into 500 mL beaker, then added 150 mL of deionized water and emulsified at 19000 rpm for 10 min under high-speed shear emulsifier. Meanwhile, a certain amount of multi-walled carbon nanotubes and sodium dodecyl benzene ring (CNTs:SDBS = 10:1) were taken in 200 mL of deionized water and ultrasonically dispersed for 30 min to obtain the multi-walled carbon nanotube dispersion. Subsequently, the CNTs were dispersed and the emulsion was mixed and stirred for 2 h under the action of mechanical stirring. Finally, the mixed solution was filtered through a filter bag to obtain the carbon nanotube-reinforced cross-linked network composites and dried in an oven at 60 °C for 1 h. Finally, composites of different CNTs/OCWPU were prepared, where the materials without additional CNTs were denoted by OCWPU, as shown in Figure 1 and Figure 2. In addition, the sample thickness of our prepared material is at 1.4 (±0.1) mm.

### 2.4. Characterization

The microstructures and morphologies of the composite films were observed by field emission scanning electron microscopy (FE-SEM, MERLIN Compact). Structural analysis of composite films was carried out using Fourier transform infrared spectra (FTIR) (Spectrum Two, Platinum Elmer Medical Diagnostic Products Co., Ltd., Shanghai, China. Electromagnetic interference (EMI) in the X-band was measured using a vector network analyzer (N5232A PNA-L, AGILENT), The composite films were placed between two X-band waveguide parts that were connected to separate ports of the VNA, the scattering parameters (S_11_ and S_12_ or S_22_ and S_21_) were obtained to calculate EMI SE. The electrical conductivity was measured with four-point probe method (RTS-8, four-point probe Technology Ltd., Guangzhou, China).

## 3. Results and Discussion

### 3.1. Morphologies of the Composites

The structure of OCNT/WPU was changed under the impact of high shear. OCNT/WPU was left to form a film in a Teflon mold and compared with OCWPU. The OCNT/WPU and OCWPU were characterized and analyzed by using infrared spectral analyzer and SEM of the cross-section. Figure 3 shows the infrared spectra of OCWPU and OCNT/WPU. It is obvious from the figure that after high shear, OCWPU increased three distinct peaks at 1990 cm^−1^, 2119 cm^−1^, 2342 cm^−1^, and other characteristic peaks were consistent with polyurethane, such as the vibrational peaks of -C=O and -N-H in carbamate at 1680 cm^−1^, 3400 cm^−1^. Among the three more obvious peaks added, a broader peak of medium intensity at 2342 cm^−1^ and two sharper peaks at 2119 cm^−1^ and 1990 cm^−1^, which may be the stretching vibration peaks of -N=C, C=C, prove that the composites material is damaged at the molecular structure level under the action of high shear emulsifier. Among them, -N=C may be caused by the destruction of the urethane group in the polyurethane molecular chain, and at the same time, the generation of C=C is caused by the grafted OCNT in the polyurethane matrix. Although OCNT contain oxygen-containing functional groups that can react with polyurethane molecules, the carbon nanotubes tend to form stress points in the polyurethane matrix due to their agglomeration. Therefore, under the action of high shear force, OCNT are more easily exposed in the matrix, and there are a lot of defects in the surface structure of OCNT, thus, the stretching vibration peak of C=C is generated.

In order to observe the comparison between OCNT/WPU and OCWPU structures more particularly, SEM was used to analyze the cross-sectional structures of both, as shown in Figure 4. Figure 4a,c show the cross-sectional structures of OCNT/WPU at different magnifications, and b and d show the cross-sectional structures of OCNT/WPU. Compared with the smooth cross section of OCNT/WPU, the cross section of OCWPU shows numerous wrinkles, as well as irregular blocky bumps. In the red squares of Figure 4c, it can be seen that there are lots of accumulations of OCNT in the bumps, and some OCNT are pulled out from the polyurethane matrix.

In contrast, there are only sporadic small bumps in the section of OCNT/WPU, which can be inferred that oxidized carbon nanotubes were tightly wrapped in the polyurethane matrix, and the agglomerated oxidized carbon nanotubes were pulled out of the polyurethane matrix under the action of external force, and finally formed small bumps. After the comparison of SEM, it is obvious that the oxidized carbon nanotubes in the waterborne polyurethane are exposed more by the high shear force. Combined with the FT-IR, it can be proved that the increased surface activity of OCWPU facilitates the adsorption of CNTs on its surface and further increases the loading of CNTs, which in turn makes the composite material have excellent electrical conductivity and electromagnetic shielding effect.

Furthermore, to investigate the dispersion of carbon nanotubes in the “broken” polyurethane and the effect on the morphological structure of the polyurethane, the cross-sectional morphology and structural characterization of CNTs/OCWPU with 2.5%, 10% and 30% CNTs loading were presented by SEM and Raman spectroscopy. At low CNTs loading, the CNTs were only agglomerated and adsorbed at some spots, and the interlocking tubular structure of CNTs could be clearly seen, and the polyurethane still maintained a large, wrinkled surface area. With the increase in CNTs content, the interior of the composite shows irregular flocculent accumulation, similar to the a, b plots in Figure 2. In Figure 5b′,c″, it can be seen that CNTs are wrapped on the surface of polyurethane, in which CNTs form an interlaced network structure and CNTs are in contact with each other, which provides a well-structured basis for the enhancement of the conductivity of the composite.

Zhao.G et al. [34] prepared IPSN structures with the help of high shear force, where the polyurethane composites showed a laminar structure in the cross section and carbon nanotubes were wrapped on the surface of the polyurethane. In their study, WPU formed a dendritic fiber structure under the action of high shear, which increased the loading of CNTs to 80% and imparted good electrical conductivity and electromagnetic shielding effect to the composites. Inspired by them, a high-shear emulsifier was used to “break” the OCWPU grafted with OCNT in the system, which exposed the OCNT in the system and helped the adsorption of CNTs at a later stage, and finally the carbon nanotube-coated composites were obtained, as shown in Figure 6.

In OCNT/WPU, OCNT were grafted into polyurethane molecular chains. Due to the structure of CNTs, they tend to form clusters, which are more prone to stress points in the polyurethane matrix. When the OCNT/WPU was subjected to high shear force, these sites of OCNT agglomeration were first destroyed, rather than the sites bound to the polyurethane matrix by chemical bonds. Due to the uneven distribution of OCNT in the polyurethane matrix, the flocculated structures formed are also of different sizes, as shown in Figure 5. These exposed OCNT are equivalent to one site, and the later added CNTs are assembled in one OCNT site in the polyurethane matrix, and at the same time, there are some intermolecular forces adsorbed on the polyurethane surface. Like the Fang Zhi cypress growing at an altitude of 3500 m in the Four Girls Mountains in Aba Prefecture, Sichuan Province, China, the surface of its branches is covered with moss all over. the OCNT in OCWPU grow in the polyurethane matrix like fungi, and with the increase in CNTs content, the fungal colony grows gradually, and the coverage area increases to form a rosette-like scale. Similarly, in Figure 5b,c, a clear tubular structure of CNTs can be seen on the flocculated surface of polyurethane, with CNTs interlocking and stacked to form a rosette-like scale. In Figure 7, two characteristic peaks of the C-atom crystal can be seen, the D-peak and the G-peak. The D-peak represents a defect in the C-atom lattice and the G-peak represents the in-plane stretching vibration of the C-atom sp2 hybridization [35,36,37]. In this work, the intensity of the D and G peaks increased with increasing CNTs content, suggesting that the CNTs in the polymer matrix have a direct effect on Raman. Therefore, calculating the two peak areas and combining them with SEM plots of CNTs/OCWPU allows the overall resultant state of the composite to be observed. The calculated ID/IG values obtained are all below 1, indicating that the carbon nanotubes maintain a relatively intact structure in the WPU system.

### 3.2. Electrical Property

In order to investigate the effect of the adsorption content of CNTs on OCWPU with the electrical conductivity of the composites, the electrical conductivity of the composites was tested using the four-probe test method, and the results are shown in Figure 8. With the increase in CNTs content, the conductivity of CNTs/OCWPU increased sharply at 5% CNTs content, similar to the results of the composites prepared from IPSN structures [34], isolated structures [38] studied previously. Subsequently, the conductivity of CNTs/OCWPU increased slowly with the increase in CNTs content until the CNTs content reached 50% would, the conductivity of 50% CNTs/OCWPU reached 5.1 S/cm. This indicates that with the increase in CNTs loading, the CNTs were randomly distributed on the surface of the polyurethane and the broken polyurethane was eventually covered by CNTs over a large area, as 5 c′ shown. The CNTs interleaved with each other to form a complete conductive pathway in the composite. In particular, the conductivity of 2.5% CNTs/OCWPU was only 0.00379 S/cm which was due to the fact that most of the CNTs were adsorbed around the OCNT exposed in the polyurethane matrix, resulting in less conductive pathways being formed and most of the polyurethane polymer structure remaining intact, as shown in Figure 5a. The conductivity of CNTs/OCWPU increases by two orders of magnitude when CNTs reach 5%, indicating that, while 2.5% of CNTs should be completely adsorbed on the grafted OCNT in the polyurethane matrix, 5% will cover a large area of polyurethane and form more conductive pathways in the polyurethane. At the same time, the increase in conductivity of CNTs/OCWPU with increasing CNTs content was again small, indicating that the increase in CNTs content did not contribute to the formation of more conductive pathways. CNTs are irregularly adsorbed in the polyurethane matrix and such a structure provides a positive basis for the electromagnetic shielding effect.

### 3.3. EMI Shielding Effectiveness

To demonstrate the potential of the CNTs/OCWPU composites for EMI shielding, the EMI shielding efficiency (*SE*) which characterizes the ability of an EMI shielding material to attenuate electromagnetic radiation were investigated over the frequency range of 8.2–12.4 Ghz. The total shielding of electromagnetic waves effectiveness (*SE_T_*) values equal to the sum of reflection loss (*SE_R_*), absorption loss (*SE_A_*) and multiple reflection loss (SEM) according to Schelkunoff’s theory [39,40], as shown in Equation (1). In addition, the types of shielding materials are determined by the *S* parameters (*S*_11_, *S*_21_, *S*_22_, *S*_12_) obtained by the vector network analyzer. According to these *S* parameters, the reflectance rate (*R*), and transmittance rate (*T*) can be calculated in the Equations (2) and (3) [41]. Generally, when *SE_T_* > 15 dB, *SE_M_* could be presented.
(1)$$SE_{T}=SE_{R}+SE_{A}+SE_{M}$$
(2)$$SE_{R}=-10\log(1-R)\;\;SE_{A}=-10\log\frac{T}{\left(1-R\right)\;}$$
(3)$$T={\left|S_{21}\right|}^{2}\;\;R={\left|S_{11}\right|}^{2}$$

*SE_T_* and *SE_R_* values at X-band frequency range (8.2–12.4 GHz) for CNTs/OCWPU with different CNTs content are displayed in Figure 9.

As shown in Figure 9a, CNTs/OCWPU with different carbon nanotube contents exhibited different EMI shielding effects. When the CNTs content was 5%, the EMI *SE* increased significantly, meanwhile, the EMI *SE* increased gradually with the increase in CNTs content, and the average value of EMI shielding effect of 50% CNTs/OCWPU can reach 66.5 dB. This trend is similar to previous studies, but the present sample exhibits a higher EMI *SE* at the same CNTs content, as shown in Table 1. As presented in Figure 9b, it is clear that both reflection and absorption contribute to EMI SE of samples. When the CNTs content is higher than 5%, the *SE_A_* of CNTs/OCWPU is considerably higher than *SE_R_*, indicating it a reflection-dominant EMI shielding material. Figure 9c shows a more distinct demonstration of the *SE_R_* of CNTs/OCWPU and the corresponding reflectance, and it can be seen that CNTs/OCWPU has a reflectance of only about 25% at CNTs content of 5% or more.

Furthermore, the gradual increase in SE_A_ is strongly related to the increasing conductivity of the material. Electromagnetic waves entering the interior of the material undergo electrical loss. On the one hand, through the electron polarization of CNTs, which leads to electrical loss due to the well conductivity of CNTs, electromagnetic waves can be converted into heat for the purpose of attenuation; on the other hand, interfacial polarization occurs at the rich interface between air, WPU and CNTs, absorbing and attenuating electromagnetic waves, as shown by the thin arrow in Figure 9d.

## 4. Conclusions

This work was undertaken to design a novel structure of CPC with CNTs as filler and an environmentally friendly waterborne polyurethane matrix, and to evaluate the effect of the “broken”- “adsorption” method on the structure and properties of CNTs/OCWPU. The most obvious finding to emerge from this study is that the high shear force breaks OCNT/WPU and generates new reactive groups, which helps the localized adsorption of CNTs in the WPU system and forms a conductive pathway, and the EMI shielding effectiveness can reach 23 dB at 5% CNTs/OCWPU and 66.5 dB at 50% CNTs/OCWPU. The results of this study support the idea that the structural design of polyurethanes is still an interesting part of the research on CPCs and that the incorporation of OCNT into the polyurethane matrix and as a breakout point is a relatively novel design. The surface activity of the broken polyurethane system increases, and the surface becomes sticky by absorbing moisture in the air easily after long time storage. So, the issue of the durability of the composite material is an intriguing subject which could be usefully explored in further research.

## Figures and Tables

**Figure 1 materials-15-06430-f001:**
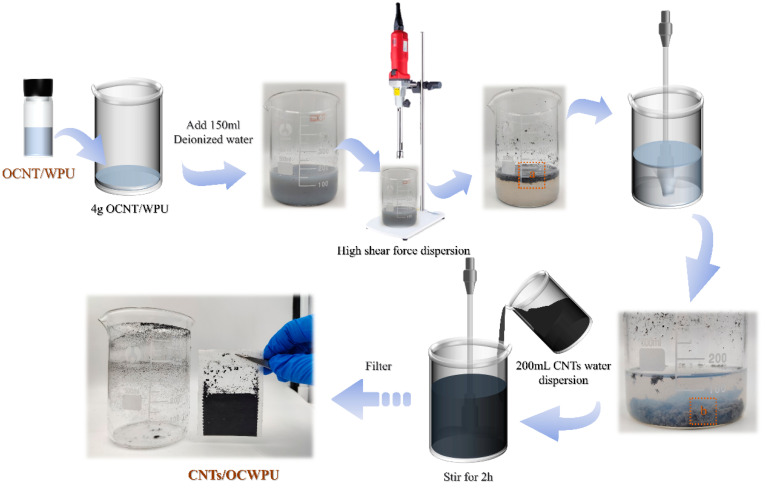
CNTs/OCWPU preparation process demonstration diagram. (a,b) as shown in Figure 2.

**Figure 2 materials-15-06430-f002:**
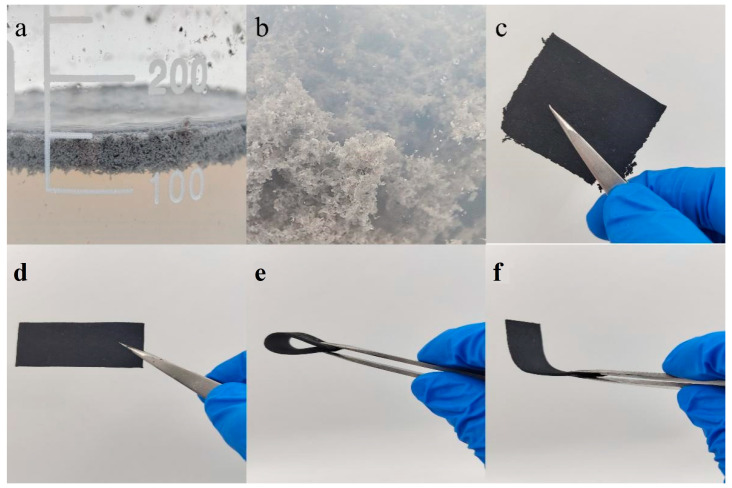
Diagram of CNTs/OCWPU preparation process. (**a**,**b**) are local enlargements in Figure 1, respectively, (**c**–**f**) are different morphologies of 10% CNTs/OCWPU after film formation).

**Figure 3 materials-15-06430-f003:**
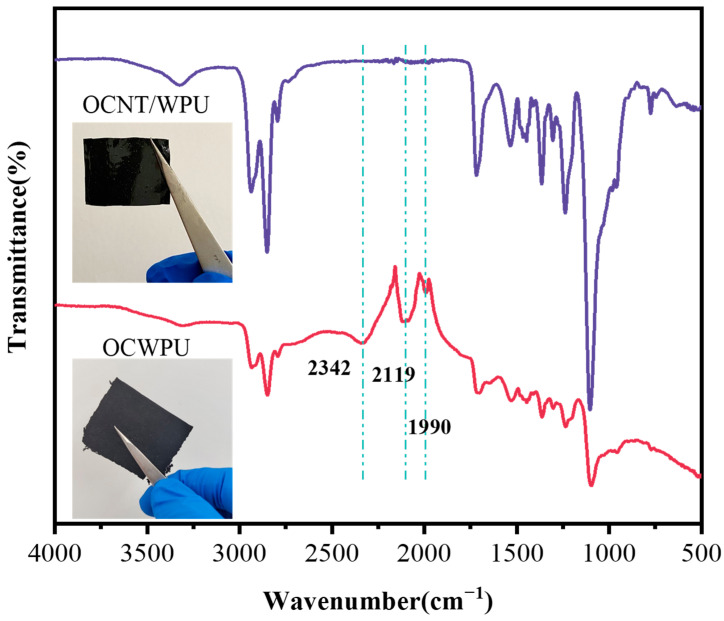
FT-IR of OCNT/WPU and OCWPU.

**Figure 4 materials-15-06430-f004:**
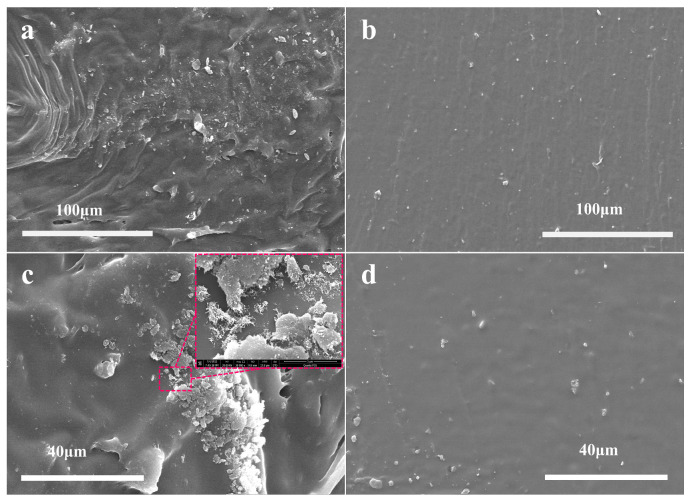
SEM images of cross-section of OCWPU and OCNT/WPU composites films: (**a**,**c**) OCWPU. (**b**,**d**) OCNT/WPU.

**Figure 5 materials-15-06430-f005:**
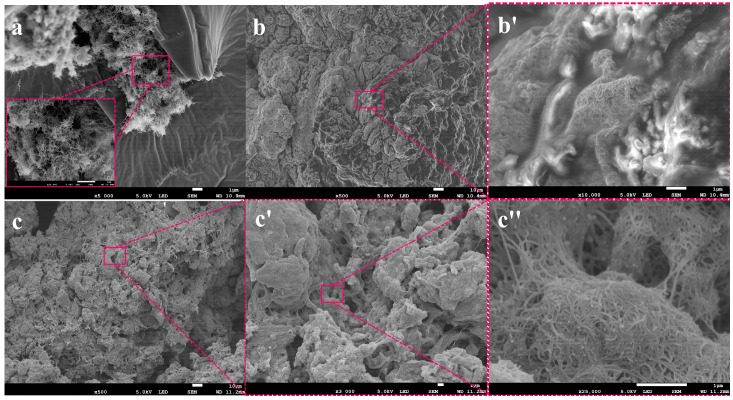
SEM images of cross-section of CNTs/OCWPU composites films with different CNTs content: (**a**) 2.5% wt%. (**b**,**b****′**) 10% wt%. (**c**,**c****′**,**c****″**) 30% wt%.

**Figure 6 materials-15-06430-f006:**
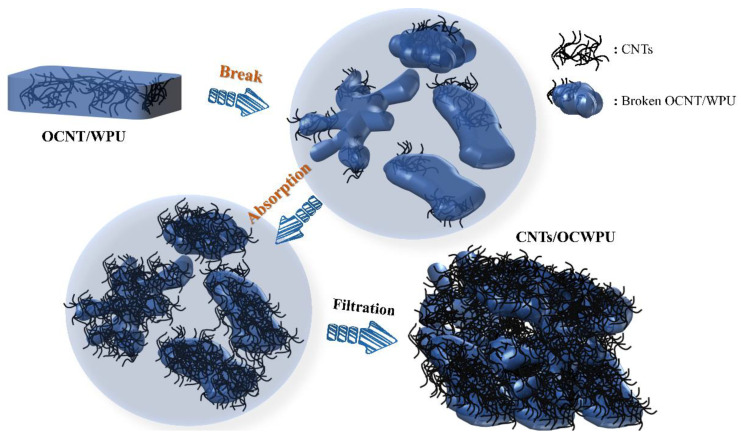
Schematic diagram of the process of CNTs/OCWPU preparation by “ broken-adsorption”.

**Figure 7 materials-15-06430-f007:**
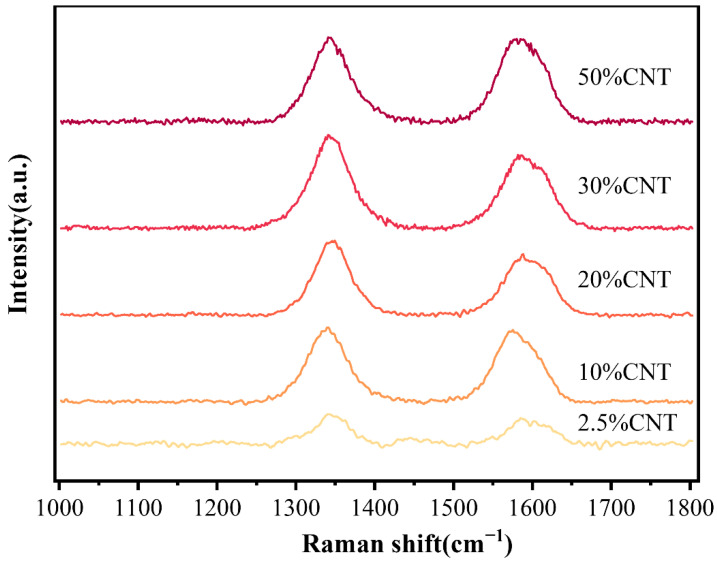
Raman spectra of CNTs/OCWPU with different CNTs content.

**Figure 8 materials-15-06430-f008:**
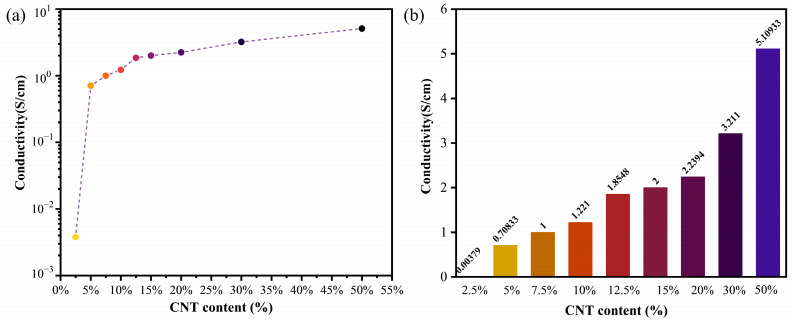
Comparative conductivity of CNTs/OCWPU with different CNTs content: (**a**) dotted line plot, (**b**) histogram.

**Figure 9 materials-15-06430-f009:**
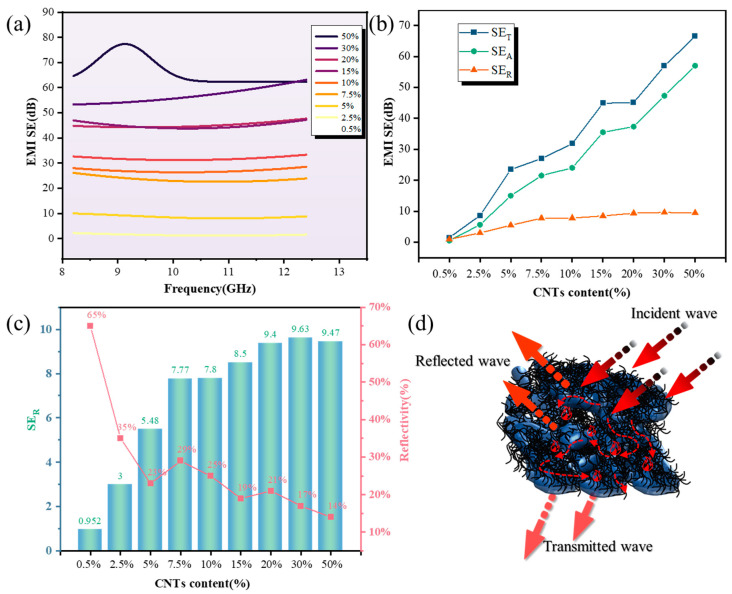
(**a**) EMI *SE_T_* of CNTs/OCWPU with various CNTs content. (**b**) Comparison of the mean values of absorption shielding (*SE_A_*), reflection shielding (*SE_R_*) and total shielding (*SE_T_*) as a function of CNTs content. (**c**) Average of *SE_R_* and reflectivity of samples at X-band. (**d**) Schematic of EMI shielding mechanism of CNTs/OCWPU composite films.

**Table 1 materials-15-06430-t001:** EMI shielding performance in the X-band of reported WPU-based composites.

Matrix	Filler	Loading (%)	EMI SE (dB)	Reference
WPU	MWCNTs	48.1	38.9	[42]
WPU	MWCNTs	40	30	[38]
WPU	MWCNTs	76.2	49.2	[30]
WPU	MWCNTs	60	40	[30]
WPU	MWCNTs	60	28.6	[43]
WPU	MWCNTs	45	25	[43]
WPU	MWCNTs	10.6	24.7	[32]
WPU	MWCNTs	10	31.8	This work
WPU	MWCNTs	30	56.9	This work

## Data Availability

Not applicable.

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
