# Peer review of "High Electromagnetic Shielding Effect of Carbon Nanotubes/Waterborne Polyurethane Composites Prepared by “Break-Adsorption” Method"

_materials, 2022, doi:10.3390/ma15186430_

Round 1

Reviewer 1 Report

Editing for grammar and clarity.  

Author Response

Dear Reviewer

Thank you very much for your pertinent comments. We checked and revised the grammar of the article, which has been noted in the text.
For the Results section, we have also made sufficient modifications. At the same time, based on the comments of other reviewers, the results section has been supplemented and revised. Therefore, the article can be displayed more clearly in front of the readers.

Reviewer 2 Report

The Present manuscript entitled ‘High electromagnetic shielding effect of carbon nanotubes/waterborne polyurethane composites prepared by "break-adsorption" method’ deals with the EMI shielding performance of CNT/PU composites. Comments are listed below

1.      Manuscript should be checked for typographical and grammatical errors.

2.      Author should maintain uniformity while writing EMI SE value. In abstract its 66.5, in text its 70, somewhere 80 dB, moreover specify at what frequency the values was obtained.

3.      Authors should also provide sample thickness, it will clear the EMISE performance.

4.      In Figure 9a shows data have quite fluctuation, it’s better to report average EMISE value for composite.

5.      Authors should also provide permittivity (dielectric study) data for more clear understanding of EMI shielding mechanism.

Author Response

Dear Reviewer

Thank you very much for your suggestions and constructive questions. We have responded to your comments and questions line by line, as follows.

  1. Manuscript should be checked for typographical and grammatical errors.

A: Thanks for your reminder, we have checked and changed the typesetting and grammar of the article, and the mistakes have been marked in the article.

  1. Author should maintain uniformity while writing EMI SE value. In abstract its 66.5, in text its 70, somewhere 80 dB, moreover specify at what frequency the values were obtained.

A: Yes, thank you very much for the reminder. We show the average value in the abstract, and in the results and discussion, we show the value of some points in the electromagnetic shielding curve in order to highlight the performance of the material, which does make the presentation of the whole article inconsistent, and we have corrected it in the article.

  1. Authors should also provide sample thickness; it will clear the EMISE performance.

A: Thank you for the suggestion, the thickness of the sample has an important effect on the electromagnetic shielding performance of the material. We did miss something in the presentation of the sample thickness. The sample thickness of our prepared material is at 1.4 (±0.1) mm, which is indicated in the Materials and Methods section of the article.

  1. In Figure 9a shows data have quite fluctuation, it’s better to report average EMISE value for composite.

A: Thank you very much for your comments, we have processed the data in Figure 9a to obtain a smoother curve and in Figure 9b we report the average EMI SE values for the composite. The two are corresponding to each other. We hope our changes will meet your requirements.

  1. Authors should also provide permittivity (dielectric study) data for more clear understanding of EMI shielding mechanism.

A: Thank you very much for the ideas. By searching the references, we know that the permittivity includes real and imaginary parts, the real part represents the storage capacity of electricity, while the imaginary part represents the dielectric dissipation. The imaginary part is mainly related to the electrical conductivity of the material, which we tested for composites, and carbon nanotubes do not contain magnetism, where dielectric losses occur mainly in composites. Therefore, the analysis of the dielectric constant does help to analyze the electromagnetic shielding mechanism of the material.

However, in this work, the type of shielding material was determined mainly by referring to the method of previous scholars, by means of S-parameters (S11, S21, S22, S12) obtained with the help of vector network analyzer. Similarly, it is found that the present material is a material of electromagnetic shielding type mainly based on absorption. Due to time, I will consider the study of the dielectric constant of the composite material in a later study. Thanks again for your suggestion!

These are our full responses, and thank you again for your constructive suggestions.

Reviewer 3 Report

The authors developed a carbon nanotubes/polyurethane composite by reacting oxidized multiwalled carbon nanotubes (MWCNTs) with waterborne polyurethane (WPU). FT-IR, SEM, and Raman were used to characterize the composite, which processes a good improvement in electromagnetic shielding effect and electrical conductivity. However, due to lack of some definition and terminology, hindered the ease to understand the significance of the work. Therefore, the reviewer would suggest reconsidering this paper after they address the following questions:

1.    Some key abbreviations are not defined in the introduction, such as OCNT/WPU, FMWCNT, EMA, PTMG, and IPDI. Some of them were introduced in the materials session. OCNT was defined twice on line 133 and line 113. What exactly do they mean?

2.    What is the method used to oxidize CNTs? The mixed acids used for oxidized MWCNTs should be specified in the methods, or the authors should cite some references.

3.    The electronic properties of CNTs greatly depend on the CNTs diameters, which can lead to different conductivity and electromagnetic shielding effects. For better repeatability and reproducibility of the material, the average diameter and distribution should be reported along with the name of the synthesis method.

4.    The authors showed FT-IR, SEM, Raman to characterize the morphology of the composite. They implied that the oxygen-containing functional groups are reacted and cross-linked into the PU without evidence/reference to show such dangling bonds on the CNTs surface can link to the reactant.

5.    The definition of carbon nanotube/waterborne polyurethane composite (OCWPU) and ??? (OCNT/WPU) are not clear in the paper. They did not even give the definition of OCNT/WPU. What is the difference between these two was not clearly stated leading to confusion to the whole concept of the paper? 

6.    What are the differences between OCWPU, OCNT/WPU, and CNTs/OCWPU? A clear and specific comparison should be provided.

7.    CNTs tend to bundle together due to a strong van der Waal interaction among each tube, therefore, solely judging from the SEM cannot draw the conclusion on line 236. Additional citations or evidence should be provided.

8.    In the Raman spectra, what is the signal from WPU? It may also contribute to the spectrum; thus blank control is necessary to be conducted. Also, the D and G peaks in Raman spectra (Figure7) are artificial, it cannot be the real Raman from MWCNTs, especially embedded in polymers.

9.  On line 277, what do you mean by carbon oxide nanotubes? The authors should have kept the name consistent and ordered.

10.   In Figure 3, the authors suggested that the high shear emulsifier may break the N=C and C=C double bonds. If such bonds break, the MW of WPU should significantly decrease. Did the authors characterize that? Also, CNTs are made of C=C double bonds and oxidized CNTs are defective, they become even easier to break and induce more defects on the surface of CNTs by such shearing force as a result, we should see D/G ratio increase. Therefore, the D/G ratio should respond to the shearing force, CNTs wt%, if their statement holds true. However, in Figure 7, the D/G ratio remains roughly the same.

11.  The author chose to use shear force to break OCWPU, would it be more efficient to use ultra-sonication (horn sonication) to do so?

12.  The electromagnetic shielding effect is interesting. In the case of shear force, what is the average length of the CNTs remaining in the composite? Since the CNTs may be cut short during the shear force, will that be a factor affecting the electromagnetic shielding effect?

Author Response

Dear Reviewer

Thank you very much for your suggestions and constructive questions. We have responded to your comments and questions line by line, as follows.

  1. Some key abbreviations are not defined in the introduction, such as OCNT/WPU, FMWCNT, EMA, PTMG, and IPDI. Some of them were introduced in the materials session. OCNT was defined twice on line 133 and line 113. What exactly do they mean?

A: Thank you very much for your reminder and please allow me to add the following: oxidized multi-walled carbon nanotubes/waterborne polyurethane composites (OCNT/WPU)、functionalized multiwalled carbon nanotubes (FMWCNT)、thylene methyl acrylate (EMA)、Isophorone diisocyanate (IPDI)。OCNT stands for oxidized multi-walled carbon nanotubes and we have updated the definition of OCNT on lines 113 and 133. Thank you very much for the reminder.

  1. What is the method used to oxidize CNTs? The mixed acids used for oxidized MWCNTs should be specified in the methods, or the authors should cite some references.

A: Thank you very much for your reminder that the preparation method of oxidized carbon nanotubes has been added in the experimental section of the paper. The details are as follows:

Took 2.5g of pristine multi-walled carbon nanotubes, 50ml of H2SO4, 5g of NaNO3 into a three-necked flask and heated the three-necked flask in a water bath to 60℃. Then, kept at a constant temperature of 60°C, stirred with a magnetic stirrer at 800r/min for 2h, followed by a centrifuge at 8000r/min for 5min to process the solution, poured out the acid solution, and repeated this operation 4 times until there was no excess acid solution in the centrifuge tube. Added the deionized water into the ultrasonic disperser for 30 min, and filtered the dispersed liquid with a vacuum pump until the pH of the filtrate was neutral. The filtered product was freeze-dried and finally ground into a powder to obtain OCNT.

Where mixed acid is a mixture of concentrated sulfuric acid and sodium nitrate.

  1. The electronic properties of CNTs greatly depend on the CNTs diameters, which can lead to different conductivity and electromagnetic shielding effects. For better repeatability and reproducibility of the material, the average diameter and distribution should be reported along with the name of the synthesis method.

A: Yes, the size of carbon nanotubes has a direct impact on the electrical conductivity and electromagnetic shielding properties of the material. Primitive multi-walled carbon nanotubes were mainly used in this study, and their specifications are shown in the Materials section. The pristine multi-walled carbon nanotubes with an average length of 20μm and an average diameter of 4nm were purchased from Chengdu Organic Chemical Co., Ltd (Chengdu, China).

  1. The authors showed FT-IR, SEM, Raman to characterize the morphology of the composite. They implied that the oxygen-containing functional groups are reacted and cross-linked into the PU without evidence/reference to show such dangling bonds on the CNTs surface can link to the reactant.

A: Yes, there may be dangling builds on the surface of CNTs, which cannot be proved to be able to cross-link with WPU substrates. However, in this article, the high activity and instability of the dangling bonds on the surface of CNTs were mainly exploited, while strong acids were used to attack the defects of CNTs to make the surface with oxygen-containing functional groups to prepare OCNT. this article also relies on the cross-linking reaction of oxygen-containing functional groups on the surface of OCNT with polyurethane, rather than the dangling bonds on the surface of CNTs. Later, the adsorption of a large amount of CNTs is also due to the special polymer structure prepared by high shear force action, which makes CNTs adsorbed on the polymer surface.

  1. The definition of carbon nanotube/waterborne polyurethane composite (OCWPU) and ??? (OCNT/WPU) are not clear in the paper. They did not even give the definition of OCNT/WPU. What is the difference between these two was not clearly stated leading to confusion to the whole concept of the paper?

A: We are sorry that the way we have presented this article has confused your understanding of it. In this case, OCNT/WPU is a composite prepared after adding OCNT to the polyurethane prepolymer stage during the synthesis of WPU. Then, we processed the prepared OCNT/WPU emulsion through a high shear force emulsifier to obtain a composite material with another structure (OCWPU). Therefore, to distinguish the structural differences between the two, we denote the material treated by the high shear emulsifier as OCWPU and the one before the treatment as OCNT/WPU.

  1. What are the differences between OCWPU, OCNT/WPU, and CNTs/OCWPU? A clear and specific comparison should be provided.

A: In order for you and the reader to more clearly distinguish the materials represented by the different abbreviations in the article, we have listed their meanings separately, as follows.

CNTs: multi-walled carbon nanotubes

OCNT: oxidized multi-walled carbon nanotubes

OCNT/WPU: composites formed by grafting oxidized multi-walled carbon nanotubes into a polyurethane matrix by in situ polymerization

OCWPU: OCNT/WPU composites were treated by a high shear emulsifier without the addition of CNTs, resulting in a new structure of the composite.

CNTs/OCWPU: Composites prepared by adding different contents of CNTs during the preparation of OCNT/WPU. (As shown in Figure 1)

In order for you to distinguish more clearly, we have readjusted the 2.3 part of the article, and we hope our answer can make you understand clearly.

  1. CNTs tend to bundle together due to a strong van der Waal interaction among each tube, therefore, solely judging from the SEM cannot draw the conclusion on line 236. Additional citations or evidence should be provided.

A: Yes, there is a strong van der Waals force between carbon nanotubes, and it can also be seen in the SEM image (Figure 5) that carbon nanotubes are entangled with each other. However, in OCNT/WPU, the surface of OCNT is with oxygen-containing functional groups (-OH, -COOH, etc.), and the -OH in these oxygen-containing functional groups can react with the -NCO in the polyurethane prepolymer, which in turn graft OCNT into WPU. Also, because OCNT grafted into the polyurethane matrix is also prone to agglomeration, it is more likely to form stress points in the polyurethane matrix and therefore more likely to be exposed from the matrix when subjected to high shear forces.

Based on your query, we found that the conclusion in the article, was not expressed clearly. Therefore, we have re-expressed it.

“In OCNT/WPU, OCNT were grafted into polyurethane molecular chains. Due to the structure of carbon nanotubes, they tend to form clusters, which are more prone to stress points in the polyurethane matrix. When the OCNT/WPU was subjected to high shear force, these sites of OCNT agglomeration were first destroyed, rather than the sites bound to the polyurethane matrix by chemical bonds. Due to the uneven distribution of oxidized carbon nanotubes in the polyurethane matrix, the flocculated structures formed are also of different sizes, as shown in Figure 5.”

  1. In the Raman spectra, what is the signal from WPU? It may also contribute to the spectrum; thus blank control is necessary to be conducted. Also, the D and G peaks in Raman spectra (Figure7) are artificial, it cannot be the real Raman from MWCNTs, especially embedded in polymers.

A: Thank you for your reminder and we agree with you that the WPU may be contributing to the spectrum in the Raman spectrum and that the D and G peaks in the Raman spectrum are also not the Raman of the real MWCNTs in the polymer. Rather, they are the characteristic Raman peaks of C atoms at 1300 cm-1 and 1580 cm-1, respectively, where the D peak represents the C-atom crystal defect and the G peak represents the in-plane stretching vibration of the C-atom sp2 hybridization.

We have probably not expressed them clearly in our discussion in the article. We performed Raman with different contents of CNTs/OCWPU, where the only variable is the loading of CNTs, in order to investigate the changes in the overall structure of the composite after the adsorption of different contents of CNTs by OCWPU. It is not very meaningful in this study if blank control is performed on WPU, because the adsorption of carbon nanotubes is performed on OCWPU in the experiment, not WPU. And OCWPU already contains a certain amount of OCNT. Therefore, what is planned in this study is to change the adsorption amount of CNTs on OCWPU, and then to observe the adsorption of different content of CNTs by Raman on the composite.

  1. On line 277, what do you mean by carbon oxide nanotubes? The authors should have kept the name consistent and ordered.

A: We are very sorry, it is our mistake, it should correctly be oxidized carbon nanotubes, and it has been corrected in the text.

  1. In Figure 3, the authors suggested that the high shear emulsifier may break the N=C and C=C double bonds. If such bonds break, the MW of WPU should significantly decrease. Did the authors characterize that? Also, CNTs are made of C=C double bonds and oxidized CNTs are defective, they become even easier to break and induce more defects on the surface of CNTs by such shearing force as a result, we should see D/G ratio increase. Therefore, the D/G ratio should respond to the shearing force, CNTs wt%, if their statement holds true. However, in Figure 7, the D/G ratio remains roughly the same.

A: I am very sorry that this is something we missed in the discussion. We agree with your opinion that the MW of WPU will be significantly reduced if the chemical bond is broken. However, we did not think of how to describe with accurate data that the MW of WPU is reduced. In the study, we can see by the apparent morphology of the polyurethane that the OCNT/WPU film without high shear action shows a smooth and bright surface, compared to the treated OCNT/WPU, while the surface is dull and rough. Moreover, before the high shear emulsifier treatment, OCNT/WPU was a light blue emulsion with a uniform texture, while after the high shear emulsifier treatment, OCNT/WPU clusters were floating in the upper layer of the emulsion in the form of fibrous flocs (as shown in Figure 1), which can also indicate that the MW of OCNT/WPU was reduced and the complete structure of polyurethane was broken, and the more complete molecular structure was broken into a dispersed structure.

Meanwhile, your mention of C=C double bond from OCNT itself defect gives us a better hint that C=C may indeed be produced by OCNTs own defect, which also fits well with our experimental results. Thank you very much for your reminder, which we will add in the discussion section of the paper.

The question about the variation of ID/IG values is because Raman in the experiment was tested on different contents of CNTs/OCWPU, where the only variable is the content of pristine multi-walled carbon nanotubes (CNTs), not the content of OCNTs. Therefore, the value of ID/IG remains basically the same, while it can be seen that the intensity of the Raman peak increases with the increase of the adsorbed CNTs content, which is in accordance with our experimental results.

  1. The author chose to use shear force to break OCWPU, would it be more efficient to use ultra-sonication (horn sonication) to do so?

A: As you said, crushing with ultrasound would be more effective. We also tried the method of breaking OCNT/WPU by ultrasonic waves. Taking the same 4g of OCNT/WPU emulsion, after the ultrasonic action, a uniformly dispersed emulsion was formed, like a complete piece of jelly, chopped into numerous small pieces. This result is indeed more homogeneous than the fibrous flocculation formed under the action of shear force.

However, it is not consistent with the purpose of our experimental study for the reason that a high shear emulsifier can generate a strong turbulent force at rotational speed, and this turbulent force can shear the OCNT/WPU into flocs formed by short fibers. As stated in the references cited in the paper, this bionanostructure can provide more adsorption space in order to increase the loading of carbon nanotubes. Also, it is in line with our research to explore new structures for conducting polymer composites.

What you said, the way of ultrasonic dispersion, we can further explore deeply at a later stage, about the specific structure as well as the properties of the polymer after ultrasonic dispersion, etc.

  1. The electromagnetic shielding effect is interesting. In the case of shear force, what is the average length of the CNTs remaining in the composite? Since the CNTs may be cut short during the shear force, will that be a factor affecting the electromagnetic shielding effect?

A: Yes, the carbon nanotubes may be sheared short under the shear force. In our experimental process, we used shear force in three places (as shown in the figure below).

The first place is when we use the high shear emulsifier, at this stage, the shear force is the largest, and the speed reaches 19000 r/min, but the main sample treated at this stage is OCNT/WPU, the purpose is to expose the OCNT in the complete emulsion by the turbulence force. At the same time, make the WPU form a fibrous flocculation to facilitate the adsorption of large amount of carbon nanotubes in the next step.

In the second place, after the high shear force, the broken OCNT/WPU floated in the upper layer of the liquid, and we stirred the emulsion with the help of a mechanical stirrer to make the floating sample evenly dispersed in the liquid for the uniform adsorption of carbon nanotubes in the later stage.

In the third place, when the carbon nanotube dispersion after ultrasonic dispersion was added to the above emulsion, we also used a low speed stirrer to stir and disperse the dispersion, and the main purpose of this stage was to let the CNTs complete the adsorption in OCNT/WPU and increase the loading of OCWPU. Throughout the preparation process, damage is caused to the CNTs, but there is no major effect on the experiment because, as can be seen by the EMI test results, there is a more significant change in the EMI value with the increase in the loading of carbon nanotubes, and also, it shows that the results prepared in this study have some improvement in improving the electromagnetic shielding performance of the composites.

For the average length of the remaining CNTs in the composite we cannot calculate it at present, but it can be seen by the SEM image of CNTs/OCWPU (Figure 5 in the paper) that the CNTs are agglomerated and wrapped around the polyurethane surface, and it can be seen that their length is not greatly damaged.

These are our full responses, and thank you again for your constructive suggestions. Hope our answer will give you a clearer understanding of the article.

Round 2

Reviewer 2 Report

The Present manuscript can be published in present form.

Author Response

Thank you very much for your approval of our revised paper! And we appreciate your very constructive comments to us!

Reviewer 3 Report

The authors have answered most of the questions and have included recommended changes. The confusion was majorly caused by the inconsistent terminology and now it has been improved. However, there is an important remaining question regarding the Raman spectra (Figure 7).  Again, all the D and G peaks in this figure are unreal. Usually, MWCNT D and G peaks should be asymmetrical, especially, G peak is commonly convoluted with D’ at ~1600 cm-1. (e.g. doi.org/10.1016/j.physrep.2004.10.006, doi.org/10.1557/jmr.2014.303, and doi.org/10.3390/fib2040295). However, Figure 7 is overly smooth and G peak is even perfectly symmetrical. The authors should provide the raw data instead of artificially fitted data.

Author Response

Dear Reviewer

We are very grateful for your approval of the revision of our article.

The Raman spectrum in Figure 7 is certainly a fitted figure that we fitted to the raw data at the time in order to calculate the ratio of the D and G peaks. We have in present updated Figure 7 in accordance with your suggestion. There can be seen that the Raman curve is definitely convoluted at 1600 cm-1. The modifications we have marked in the article.

Thanks again for your comments!

Round 3

Reviewer 3 Report

The authors have made the suggested revisions, thus I do recommend publishing this work.